# Modeling the Directed Evolution of Broad Host Range Phages

**DOI:** 10.3390/antibiotics11121709

**Published:** 2022-11-27

**Authors:** James J. Bull, Holly A. Wichman, Stephen M. Krone

**Affiliations:** 1Department of Biological Sciences, University of Idaho, Moscow, ID 83844, USA; 2Institute for Modeling Collaboration and Innovation, University of Idaho, Moscow, ID 83844, USA; 3Department of Mathematics and Statistical Science, University of Idaho, Moscow, ID 83844, USA

**Keywords:** phage therapy, natural selection, recombination, theory, computation, ordinary differential equation

## Abstract

**Background:** The host ranges of individual phages tend to be narrow, yet many applications of phages would benefit from expanded host ranges. Empirical methods have been developed to direct the evolution of phages to attack new strains, but the methods have not been evaluated or compared for their consequences. In particular, how do different methods favor generalist (broad host range) phages over specialist phages? All methods involve exposing phages to two or more novel bacterial strains, but the methods differ in the order in which those hosts are presented through time: Parallel presentation, Sequential presentation, and Mixed presentation. **Methods:** We use a combination of simple analytical methods and numerical analyses to study the effect of these different protocols on the selection of generalist versus specialist phages. **Results:** The three presentation protocols have profoundly different consequences for the evolution of generalist versus specialist phages. Sequential presentation favors generalists almost to the exclusion of specialists, whereas Parallel presentation does the least so. However, other protocol attributes (the nature of dilution between transfers of phages to new cultures) also have effects on selection and phage maintenance. It is also noted that protocols can be designed to enhance recombination to augment evolution and to reduce stochastic loss of newly arisen mutants.

## 1. Introduction

Collectively, phages are the main global predators of bacteria. However, individual phages tend to have narrow host ranges, often not even encompassing entire bacterial species; even when phages are found to infect taxonomically diverse bacteria, they typically do not infect broadly within those bacterial groups [1,2,3,4,5], and the diversity of bacterial anti-phage defense mechanisms virtually ensures an absence of systematically broad host ranges [6].

When phages are employed to kill bacteria, as in phage therapy, broad host ranges are desirable. A century ago, phage therapy was used as a method of curing bacterial infections [7], but it was abandoned in Western medicine when broad-spectrum antibiotics were developed [8,9,10]. One of the possible reasons for its demise may have been the narrow host range of phages, because phages were employed against infecting bacterial strains that were not necessarily susceptible.

Now, with the rise of antibiotic-resistant bacteria, phages are being reconsidered as a treatment for infections. A phage with a broad host range facilitates treatment of many patients using a single formulation. The goal may be to have few phages with individually broad host ranges or a cocktail of several phages with a collective, broad host range. As most phages isolated directly from nature lack the desired host range for any application, protocols have been developed to artificially evolve phages to attack the bacteria of interest [11,12,13]. There are likewise cases in which it has been difficult to find phages for treating a patient’s infection, for which a method to rapidly evolve phages to attack a bacterial strain would be useful [14,15,16]. Furthermore, bacteria may quickly evolve resistance to specific phages, rendering treatment with single, narrow host-range phages of temporary benefit [14,17]. An easy method for evolving new host ranges would allow the clinician to choose from a wide selection of phage characteristics that may be desired, in essence choosing the ideal phage first (e.g., for blocking host resistance, fast growth, low immunogenicity, use of specific bacterial receptors) and then evolving its host range to attack the strains of interest.

Protocols for host range expansion have been developed empirically, with little attention to the quantitative details that may affect a method’s success. Part of the reason for this inattention to detail is pragmatism: if a method yields an acceptable outcome, then there is no practical need to understand why. Yet, as phage therapy looms ever larger as an alternative to antibiotics, cases are arising in which phages are limiting—there may be no phages to initiate treatment, or phages may be lacking for treatment of bacteria that evolved to resist the first round of phage therapy. This increased applicability of phage therapy warrants understanding the various methods that are available to expand its utility.

Our focus is on the ‘directed’ evolution of host range in phages—the use of a laboratory environment to evolve one or more phages to infect several host strains. We apply mathematical/computational models to compare some of the different protocols that have been developed. This approach is not a substitute for empirical work, but it nonetheless provides a relatively straightforward comparison of the host range characteristics likely to be evolved in the different protocols and provides a foundation for hypotheses. The problem is intrinsically simple when the goal is to obtain phages growing a single (new) host. However, once the goal expands to phages that grow on two or more new hosts, the possible outcomes range from ‘specialist’ phages that each grow on single hosts to ‘generalist’ phages that grow on multiple hosts. The protocol influences these outcomes.

Host-range expansion protocols can differ in many ways. The methods so far developed for growth on several host strains are similar in that they expose phages to the different host strains to specifically amplify the phages that grow. However, three methods have been used that differ in how those multiple hosts are presented to phages (Figure 1):Parallel presentation. Phages are exposed to each host in separate cultures, with the phages (culture supernatant) from all cultures pooled free of cells and again distributed to the different hosts [e.g., the Appelmans protocol and variations of it, [18,19].Sequential presentation. Phages are grown on one host, then the pool of phages from that host is transferred to another host, and so on [20]. The process continues by rotating through each host and is then repeated. In one variation of this protocol, actual plaques are chosen for propagation from one host to the next [20].Mixed presentation. Phages are grown in a mix of all hosts in a single culture. The bacterial mix is periodically or continually refreshed to ensure the continued presence of all hosts while maintaining the supernatant containing phage. This method originates with d’Herelle [7] using one non-permissive and one permissive host [as attributed by [11], p. 129].

Any of these presentation protocols may be used with few or many hosts, and the degree of bacterial strain diversity may be varied as well. Typically, all methods include exposure to a permissive host to ensure the continued maintenance of the phage and include one or more initially non-permissive hosts on which phage growth is desired. Protocols may likewise differ in the number and diversity of phages used at the start.

Perhaps the most common approach to the directed evolution of host range has been to screen for plaques on the non-permissive host [11,20,21,22]. The host presentation protocols instead propagate phages without relying on plaque formation. One problem with plaque isolation as the sole basis of selection is that poorly growing phages may not form plaques initially, but they may quickly evolve to do so. The blind transfer of a supernatant containing poorly growing phages facilitates the evolution of better phage growth, so that phages can eventually be obtained that do form plaques.

The focus of this study is to evaluate host presentation as it affects the directed evolution of host range. That is, how does the host-presentation protocol affect selection of generalists over specialists—phages that grow on one host versus many? The empirical goal may be to generate a single, broad host range phage, or instead to generate phages that grow best on any of several hosts, and different protocols will have different outcomes toward these goals. However, evolution also depends on the introduction of (favorable) variation and the maintenance of that variation long enough for selection to elevate its frequency. Following the treatment of selection, the Discussion will address some features of protocols that affect these other contributions to evolution.

## 2. Methods

### 2.1. Growth Dynamics

Each of the three presentation protocols alternates a phage growth phase with a sampling/dilution/transfer phase. This cycle is repeated until either the desired outcome is achieved or some total duration of effort has been reached. We modeled the growth phase as a system of ordinary differential equations assuming one or two hosts and three phages. We assumed lytic phage growth with constant rate parameters and exponential growth of bacteria. Following [23], the general framework for our differential equations (which can be expanded to an arbitrary number of phages and hosts) is:(1a)$$\frac{dP_{i|j}}{dt}=b_{i|j}\lambda_{ij}I_{ij}-k_{ij}P_{i|j}C_{j}$$
(1b)$$\frac{dI_{ij}}{dt}=k_{ij}P_{i|j}C_{j}-\lambda_{ij}I_{ij}\;\;$$
(1c)$$\frac{dC_{j}}{dt}=r_{j}C_{j}-\underset{i}{\sum }k_{ij}P_{i|j}C_{j}\;$$

Here, i∈{A,B,AB} is the index for the phage strains and j∈{A,B,(A,B)} is the index for the bacterial environments into which the phage are placed, with (A,B) denoting the mixed environment with both hosts. $P_{i|j}$ is the density of free phage *i* growing in bacterial environment *j*; $I_{ij}$ is the density of strain *j* cells infected by strain *i* phage; $C_{j}$ is the density of bacterial strain *j* (Table 1). The lysis rate for the $I_{ij}$ infected cells is $\lambda_{ij}$, and $b_{i|j}$ is the corresponding burst size; the adsorption rate for phage strain *i* onto bacterial strain *j* is $k_{ij}$ and $r_{j}$ is the intrinsic growth rate of bacterial strain *j*. Aside from burst size, which we model as depending on the phage strain, we took the parameters to be constant even though their dependence on *i* and *j* in practice is understood. The sum in (1c) is over all phage strains *i* that are placed in a single culture of bacterial strain *j*.

For parallel host presentation, there are two separate bacterial cultures j=A,B into which we place all three phage strains, resulting in phage densities $P_{i|A}$ and $P_{i|B}$ (that are then combined to get density $P_{i|A}+P_{i|B})$. For sequential host presentation, we have one bacterial strain at a time, *j = A* or *j = B* depending on which phase of the alternating cycle we are in. For mixed host presentation, j=(A,B).

Trials followed these equations for 20 time-step ‘cycles,’ at which point the phages from cultures were recovered, combined (for Parallel presentation), diluted, and mixed with cells at a renewed cell density of 10^7^ (with Mixed presentation, both hosts were renewed at 10^7^). These new cultures were again assumed to proceed for another 20 time-step cycle, whence phages were recovered, and so on. Beginning with equal densities of all three phage types, the process ended after 1000 time steps (50 cycles). Differences in host presentation protocols were reflected in the equations. Thus, for Parallel presentation, a separate set of equations (accommodating one cell type and three phages) was used for each host. For Mixed presentation, the same, single set of equations was used for each cycle (two cell types, three phages in a single culture), whereas Sequential presentation used a single set of equations for three phages with one cell type, but the cell type changed between cycles.

Except for burst sizes and bacterial growth rates, the same set of parameter values was used for all trails and for all (*i*,*j*) (Table 1). Omitting subscripts, *r* = {0.1, 0.3}, λ = 1.0, and *k* = 10^−9^. If a time step is equated to an hour, these values approximate what might be used in the lab for phages from nature, but they do not span the spectrum of what is known [24]. Thus, a cycle would correspond to just under a day, a trial of 1000 time steps would correspond to just under two months. Bacterial growth rates would be slower than is typical for *E. coli* in liquid culture but within the realm of other bacteria under less ideal conditions. An average lysis time of 1 hr would be long for some phages, but the exponential decay stemming from ordinary differential equations results in a highly asymmetric distribution of lysis intervals, and it is known that early lysis has a disproportionate effect on phage growth when the phage population is increasing [25]. Additionally, although these equations characterize phages with 3 parameters (burst size, lysis time, adsorption rate), trials varied only burst size. We consider that use of this single variable is adequate to capture other causes of differences in phage growth rate, and in turn that phage growth rate is the determinant of competition outcomes in these models.

We acknowledge that any implementation will violate some of our assumptions in ways likely to affect the outcomes quantitatively: phages will have discrete lysis times, bacterial suitability for phage growth will change over the course of the culture duration, bacteria will diversify into different structural states (planktonic and wall growth) that are differentially susceptible to phages, and resistant bacterial will begin to evolve in the cultures. The models are thus intended to capture broad features of protocol effects on phage competition, but individual cases will certainly be affected by details. Furthermore, the major goal of this study is to compare presentation protocols for their impact on the evolution of generalists over specialists, and that comparison should be robust to parameterizations.

Numerical analyses of these equations were carried out both using C code written by us and, in many cases also with Mathematica^®^ 13.1 using NDSolve. Individual trials were run for 1000 time steps. The C code used the Euler method with a step size of 0.001. The two approaches yielded indistinguishable patterns of evolution; all heat map figures were derived from the C codes with graphics using ggplot2 [26].

### 2.2. Dilution Protocols

Any host-presentation protocol will involve dilution of the phage. Dilutions were simulated to occur at the end of every ‘cycle’ (defined as 20 time steps). The specific order of events was that phages were recovered, pooled (for Parallel presentation), then diluted when being mixed with hosts at a renewed cell density of 10^7^ to start a new cycle. With Parallel presentation, the different host cultures were assumed to have equal volumes, so that a phage’s density on one host was combined with its density on the other host weighted equally. One key assumption was that infections and free phages had different fates at the end of a cycle. To mimic how actual protocols might operate to recover phages free of bacteria (e.g., chloroform treatment or filtering), we assumed that all infected states were lost; this assumption often had little consequence, as cells had been exhausted and infected states minimized by cycle’s end. Trials were also run in which the infected states were carried over to the new cultures (at the same dilutions as free phages), with no fundamental change in patterns.

Two dilution protocols were evaluated, denoted fixed volume and fixed count (Figure 1). The fixed-volume dilution was a simple transfer of 5% of the (free) phage density into the new cultures; with Parallel presentation, 5% of phage densities from each culture went into each of the new cultures. In contrast, fixed-count dilution reduced the phage density to 1000 in the new cultures (with Parallel presentation, 1000 phage were introduced into each of the new cultures). As phage densities at the end of a cycle often reached 10^8^ or more, the two dilution protocols often resulted in orders of magnitude more phages being transferred in the fixed-volume dilution method than in the fixed-count method. Thus the two methods differ both in density of phages transferred and in the fraction of phage transferred from the previous cycle, but these dilution-protocol differences would also have other effects in any empirical system.

### 2.3. A Historical Note

An expansive Parallel presentation method referenced below is called the Appelmans protocol. We consider the Appelmans protocol to be the method described by Burrowes, Molineux, and Fralick [18] and fundamentally similar to that of Mapes et al. [19], who credit their method to the dissertation of Burrowes. The paper usually cited for the Appelmans protocol [27] merely describes serial dilution as a way of determining phage concentration; there is no mention of multiple phages or non-permissive hosts. Burrowes et al. [18] attribute their method to the Eliava Phage Institute in Tblisi, Georgia.

## 3. Results

Our primary focus is the effect of protocol on selection among phages with different growth properties. The basic protocol is that phages are mixed with host cells, grown 20 time steps (a ‘cycle’), diluted and redistributed to new hosts. Several simplifications are used to render the analysis both interpretable and general: (1) all phage types are present and common enough to have escaped random loss: deterministic selection is operating; (2) all trials start with one generalist phage and two specialist phages; (3) two hosts are used, either together in one culture (Mixed), separately but concurrently (Parallel) or temporally separated (Sequential). The two hosts are designated A and B; the three phages are ϕ_A_, ϕ_B,_ ϕ_AB_ the subscript indicating which hosts are permissive to each phage. Thus ϕ_A_ is a specialist on A, ϕ_AB_ is the generalist that grows on both hosts.

Despite these simplifications, the results should generalize. For example, since it is apparent that the best-growing specialist using a host will prevail over other, ‘lesser’ specialists using the same host, the restriction to a single specialist per host is reasonable. The use of two hosts seems the only practical way of developing intuition; there can be little doubt that results will generalize in many ways beyond two hosts, but the combinatorial possibilities arising from more than two hosts is quickly cumbersome. By assuming deterministic dynamics, the models become agnostic to the origins of the different phages. It thus makes no difference whether the ‘mutants’ arose via mutation and recombination in the course of phage growth in the laboratory, were isolated from the wild and combined into a pool that is being grown, or are the products of an engineered library [28,29].

We use two approaches in addressing questions about selection. First, we develop a simple heuristic approach that requires almost no analysis. These methods strip the selective process to the bare minimum, reducing phage growth to a single number per culture. Despite its simplicity, this approach is informative. Beyond that we use formal dynamic calculations of each host presentation protocol, with some variations.

### 3.1. Heuristic Models

#### 3.1.1. Parallel Presentation

We begin with an approach that omits many details of phage growth but may help deconvolute the complexity of comparisons between a generalist and specialist phages. The first example assumes Parallel presentation: the phage pool is distributed to separate cultures of host A and B, grown, then pooled again before the next exposure to hosts. Phage growth in a culture is assigned a single number that reflects the number of descendants per phage at the end of the culture period, immediately before dilution (Table 2).

The subscript of *N_i|j_* gives the designation of the phage (*i*) followed by the host on which it is grown (*j*). A value of 1 is assigned to each of the specialists in cultures of their respective non-permissive hosts to represent phage survival without growth.

After one round of growth, the numbers of phages in the pool combining A and B cultures will have changed in proportion to the numbers given in Table 3:

The ratios of the A-specialist to the generalist have changed by
(2a)$$\frac{N_{A|A}+1}{N_{AB|A}+N_{AB|B}}$$

Similarly, the B-specialist has fared against the generalist according to
(2b)$$\frac{N_{B|B}+1}{N_{AB|A}+N_{AB|B}}$$

If either ratio exceeds 1, the specialist on that host will be increasing over the generalist, and vice versa.

These results reveal that, with Parallel presentation, a generalist has an arithmetic advantage over specialists from its growth on multiple hosts. However, this advantage is modest such that a relatively meagre superiority of the specialist will allow it to avoid being displaced by the generalist. There is an asymmetry favoring the generalist: whereas the superiority of the generalist on one host ensures the loss of that specialist, a superiority of the specialist on one host does not ensure loss of the generalist because the generalist may be maintained via its growth on the other host. There are three possible outcomes. (1) the generalist has a selective advantage over and thus displaces both specialists. (2) the two specialists each have advantages over and displace the generalist. (3) the generalist has an advantage over and displaces one specialist but not both. If the goal is to obtain a generalist phage, the second outcome is the worst because the generalist will be lost. Even if the generalist displaces only one of the specialists, and thus is not the only phage in the evolved culture, it can be recovered from the selection by testing individual phages.

The simplicity of these numbers belies a complexity. Specifically, the *N_A|A_* and *N_AB|A_* are not independent: rapid growth of one phage on A will affect host abundance and thus affect the numbers attained by the other phage on A (likewise for *N_B|B_* and *N_AB|B_*). Yet, this interdependence will matter only if the culture is grown long enough that bacterial numbers are overwhelmed by phage growth. Additionally, the *N_i|j_* are not necessarily constant across cycles, depending on the numbers of phages added to a culture (which may differ across cycles). These simple numbers are therefore offered merely as a heuristic. Furthermore, the numbers *N_i|j_* cannot be analytically calculated from easily measured phage properties such as burst size, lysis time and absorption rate. Indeed, if the bacteria grow rapidly enough relative to the phage, a phage with a low growth rate on one host may, in time, actually attain a higher *N_i|j_* than if it had a high growth rate [30].

The ratios in Equation (2a,b) suggest a possible undesirable outcome of Parallel presentation, what might be called a ‘tyranny’ of generalists on good hosts. Suppose that, regardless of phage, host A is very productive for phage infection but B is not (*N_A|A_* and *N_AB|A_* are large, *N_B|B_* and *N_AB|B_* are small). The ratios indicate that, even if the specialist on B is much better than the generalist on B, the generalist may displace that specialist because of the generalist’s high output on A. If host B is an important target for phage therapy despite its unsuitability for phage growth, a Parallel presentation protocol will work against the evolution of specialist phages that excel on B. Below, we suggest a modification of Parallel presentation to avoid this outcome.

A further complication is that these ratios reflect the relative fates of the different phages but not whether they grow fast enough to be maintained. Maintenance will depend on both selection and dilution. In the trivial case, the phage pool could be diluted so much that all phages vanish—the dilution across transfers exceeds the largest *N_i|j_.* Less obvious, strong differences in phage production between hosts A and B combined with extreme dilutions can result in the phage pool purging any specialist growing on the poor host for demographic reasons, not because of its evolutionary inferiority. The specialist on the poor host could have a selective advantage over the generalist on that host, yet be lost because it is diluted too much, while the generalist is maintained because of its growth on the good host. Such a case will be illustrated below.

#### 3.1.2. Sequential Presentation

Here, the entire phage pool is cultured on one host (A), whence the resulting phage pool (cleared of bacteria) is grown on host B, and the cycle is repeated. Yu et al. [20] proposed two versions of a sequential protocol, and they specifically suggested that sequential was a good method for evolving generalists. One method (their Method A) selected plaques on one host to continue the transfer to the next host. The other method (B) involved the transfer of supernatants from host to host. Method A could not possibly retain specialists or generalists that did not form visible plaques. Our focus is therefore on Method B. Using our heuristic approach, phage growth across one round of alternation will be proportional to the following (Table 4):

With relative performances:(3a)$$\frac{N_{A|A}}{N_{AB|A}\;\cdot \;N_{AB|B}},$$
(3b)$$\frac{N_{B|B}}{N_{AB|A}\;\cdot \;N_{AB|B}}$$

It is now easily seen that, in contrast to Parallel presentation, the generalist grown under alternating hosts realizes a multiplicative/geometric advantage from its growth on each host; the specialists grow only on their permissive hosts. This should strongly favor the generalist.

#### 3.1.3. Mixed Presentation

This protocol is easy to implement but offers the greatest challenge for intuition. All three phages are growing in one culture of both hosts. In contrast to the other two protocols, the generalist quantities N_AB|A_ and N_AB|B_ cannot be separated from each other because the generalist is infecting both hosts in the same culture: its numbers from one host affect its numbers on the other host. To create a combined value for the generalist merely trivializes the problem to the point of uselessness. Some of these challenges have been addressed in models that assume a constant abundance of hosts [31], but the protocols for host range evolution often violate this assumption. We thus rely entirely on numerical analyses for Mixed presentation.

### 3.2. Numerical Analyses

To provide a more quantitative sense of selection, we numerically analyze ordinary differential equations similar to those in Equation (1a,b) that encode phage growth, bacterial growth, and bacterial death from phage attack (see Methods for the structure of equations, see Data Availability for access to the files). Our approach is to compare the different host-presentation protocols, but it should be appreciated that, even with just two hosts, many variations are possible within one host-presentation protocol, affecting the dilution at the end of a cycle, the duration of a cycle, initial conditions, and bacterial growth. Our approach is to consider a limited set of conditions for one presentation protocol, then to vary some of those conditions to measure the impact on the types of phages retained. A systematic analysis that varies many conditions together would be unmanageable. The intent is to develop a sense of what set of conditions an empiricist might use to steer the outcome toward generalists or specialists, but any empirical application will likely benefit from numerical analyses tailored specifically to it.

#### 3.2.1. Parallel Presentation

Recall that, with Parallel presentation, phages are grown on each host separately and then pooled for the next cycle. Each panel in Figure 2 is a heat map illustrating the fate of one of the three phages under Parallel presentation with a fixed-count dilution (the phage pool was diluted to a density of 1000 every cycle). For a single pair of specialist burst sizes, the figure shows whether the generalist or specialists are maintained across a range of generalist burst sizes. Note that phage maintenance will generally depend both on evolution and demography, but for these trials, the outcomes are due entirely to evolution.

The results are clear (Figure 2) and in agreement with the impressions from the heuristic results in Equation (2a,b). The fate of the generalist is governed by its burst sizes relative to the coordinates of the pair of specialist burst sizes (*b_A|A_* =15, *b_B|B_* = 15, given by the coordinates of the black-on-white ring): the generalist is lost if neither of its burst sizes is as good as that of the respective specialist. Likewise a specialist is lost if the generalist has a superior burst on that host. There are intermediate zones, and the generalist is not completely lost until its burst sizes fall somewhat below those of the specialists. Each specialist is lost as the respective generalist burst size exceeds 15, but there is a slight ‘shoulder’ effect in which the specialist is more prone to loss as the generalist performs well on the other host. This latter effect is consistent with the ratios in Equation (2a,b), that the generalist is somewhat better at suppressing one specialist as it does relatively better against the other specialist. Overall, Parallel presentation is a ‘fair arbiter’ of phage performance on each host: the best-growing phages are retained on each host.

For the parameter values considered, the generalist is retained in approximately ¾ of the space shown. The space shown does not necessarily reflect the spectrum of biological possibilities, of course. If the generalist was constrained so that it could not outperform specialists on each host, for example, the relevant space would be limited to the lower left quadrant, for which the generalist would be mostly absent.

Now consider an asymmetric case when the specialists have different burst sizes (*b_A|A_* = 20, *b_B|B_* = 10, Figure 3). This case could arise if one host is better for phage growth than the other, or just if one specialist happens to be poor at growth. The pattern for the generalist from Figure 2 is shifted to the new specialist coordinates but is otherwise largely the same as before: the generalist is maintained if its burst size on at least one host exceeds that of the specialist for that host. Again, the heuristic results in Equation (2a,b) agree with the numerical outcomes.

We next consider how these outcomes depend on details within Parallel presentation (a summary Table of the different cases is offered in Section 3.3). The trials in Figure 2 and Figure 3 assumed a bacterial growth rate of *r* = 0.3 and a fixed-count dilution to density 1000 every cycle. As phage density often reached 10^9^ or more, this dilution allowed phage growth by many logs every cycle. Those analyses were modified in either or both of two ways: (i) bacterial growth rate was reduced to 0.1, and (ii) the pool was diluted by a fixed volume (5% of the phage density). The time between phage pools remained the same as before, at 20 steps. Only the asymmetric case was considered (*b_A|A_* = 20, *b_B|B_* = 10).

Merely lowering the bacterial growth rate (with no change in the dilution protocol) has a large effect on the outcome: only the A-specialist is retained in much of the space, and its effect at purging the generalist is symmetric despite the asymmetry in specialist burst sizes (Figure 4). That is, despite the presence of two phages that can grow on B, both are lost in much of the space, and the B-specialist is lost everywhere. This radical change is necessarily due to the reduction in bacterial growth rate because that is the only difference from the previous analysis.

On first impression, this fundamental change in outcome is wholly unintuitive. Merely reducing bacterial growth rate is expected to allow faster clearance by the phage because there are fewer bacteria in the culture. The observed effect is the opposite: none of the phages grows fast enough to exhaust its host by cycle’s end. The two panels of Figure 5 contrast phage growth dynamics between the *r* = 0.1 and *r* = 0.3 cases (**A** and **B**, respectively), where it is easily seen that the lower bacterial growth prevents phages from exhausting their hosts by cycle’s end. With this change in phage growth dynamics combined with the use of a fixed-count dilution, the fastest-growing phage in the pool (whether that phage grows on just host A or B or both) sets the dilution limit, and all other phages are diluted more than they grow. Hence, all other phages progressively disappear. Thus, once the burst size of the generalist exceeds 20—on either host—it now displaces the A-specialist for the same reason the A-specialist was displacing the other phages when it had the superior burst. Thus the effect is symmetric despite the asymmetry in specialist burst sizes.

This radical change in outcome is due to demographic effects of the protocol rather than to evolution. Ironically, reducing bacterial growth rate limits phage growth and thereby also prevents slow-growing phages from exhausting their hosts in 20 time steps. Phage growth depends on the product *P·C*, so phage growth increases with bacterial density and more quickly exhausts the cells at high cell density. With lower bacterial growth, phage numbers never get high enough to exhaust cells by cycle’s end. By this logic, the problem should be reversible in various ways. Indeed, the effect can be reversed without changing bacterial growth rate. For example, if the cycle length s increased, eventually the fastest-growing phage runs out of hosts, allowing slower-growing phages (on the other host) to exhaust their host and catch up to the dilution limit. Thus, increasing the cycle to 40 steps while maintaining *r* = 0.1 restores the pattern approximately to that of Figure 3.

This case reveals that non-independence of phage growth on different hosts can arise through the dilution protocol (also dependent on culture conditions). This outcome is not due to a change in the relative advantage of one phage over another (i.e., not due to selection and thus not reflected in the ratios of Equation (2a,b)) but rather stems from the demographic consequences of the dilution protocol. Phages can be lost because they cannot grow fast enough to maintain themselves, even though they may be ‘evolutionarily’ superior to their competitors who are also lost. The host-presentation protocol thus has ramifications for evolution and, separately, for demography.

The second modification of Parallel presentation considered here is to change the dilution mode while retaining the low bacterial growth rate of 0.1. Previously, with fixed-count dilution and low bacterial growth rate (Figure 4 and Figure 5A), the phage density reached just over 10^6^ by cycle’s end, so the dilution to 1000 phage was a 10^−3^-fold reduction in phage density. Phage density could not get high because the starting phage density was always the same number, and bacterial densities did not support phage growth rapid enough to overwhelm the culture. With the fixed-volume dilution, each new cycle is started with 5% of the phage density in the phage pool. This means that any gains in total phage density in one cycle directly increase the density of phages transferred into the next cycle, which in turn results in even more phage at the end of the next cycle, and so on. This has the potential for phage concentration to become so high that bacterial density becomes limiting before the cycle’s end. Indeed, whereas phage density under fixed-count dilution ultimately reached and was maintained at just over 10^6^ with transfers of 1000 phage, phage density reached almost 10^9^ when transferring 5% of the pool, at which point hosts became limiting. When bacterial density becomes limiting, phages from host A no longer drive the demographic extinction of phages on host B. The patterns with fixed-volume dilution and low bacterial growth are now closer to those of fixed-count dilution with high bacterial growth (Figure 6), although there are now also broad intermediate zones.

Although the use of a fixed-volume dilution protocol (combined with low bacterial growth) largely avoids the demographic purging of phages and restores the pattern of phage retention seen in Figure 3, the patterns in Figure 3 and Figure 6 have a striking difference: the generalist drives specialist extinction over broader parameter ranges in Figure 6. In Figure 6, the generalist is extinguishing specialists that are decidedly superior on the respective host. The reason for this change in outcome is not immediately clear. We conjecture that the difference lies in the lesser phage growth per cycle in Figure 6 due to a combination of low bacterial growth rate and fixed-volume dilution. Thus, the dilution in Figure 3 allowed approximately 6 orders of magnitude phage growth per cycle, whereas the dilution in Figure 6 allows approximately 20-fold phage growth per cycle. This reduced growth per cycle has the effect of reducing the impact of burst size differences on the *N_i|j_* in the heuristic Equation (2a,b). With smaller *N_i|j_*, the arithmetic advantage of the generalist looms ever larger.

This change in outcome (from fixed-count dilution to fixed-volume dilution, both with low *r*) is readily seen to stem from a restoration of favorable demography—phages are no longer being diluted to extinction. Avoiding demographic extinction is merely a matter of growing the phages long enough relative to dilution, an outcome that can be achieved either by longer growth or lesser dilution. As noted above, there are multiple ways to achieve this outcome, and an important one is presented next.

These few variations on the Parallel presentation protocol point to manifold complexity stemming from protocol details such as cycle duration, bacterial growth rate and dilution. They likewise motivate ways to avoid ‘unfair’ recovery of phages. One modification that could ensure maintenance of phages on each host despite differences in bacterial growth rate is to set the dilution separately for each host and then pool the samples. This protocol would also mitigate a tyranny from the generalist and prevent phage loss due to poor growth on one host even with high dilution rates. Numerical analyses support this intuition (Figure 7 for comparison to Figure 4).

#### 3.2.2. Sequential Presentation

The advantage of a generalist over specialists under sequential presentation suggested by the heuristic analysis in Equation (3a,b) is equally evident from numerical analyses (Figure 8). The Sequential protocol strongly favors the generalist even to the point of displacing far superior specialist phages. The heuristic analysis in Equation (3a,b) provides ready intuition for why this is so: the generalist has a multiplicative advantage over specialists from its growth on both hosts. Poor growth on two hosts can exceed good growth on a single host.

Alternating hosts does not change phage properties per se, it merely selects a wider set of broad host range phages than does parallel presentation. Perhaps surprisingly, many of the issues with Parallel presentation are still present with sequential presentation. Thus, depending on dilution rates and bacterial growth, it’s possible to purge all phages that grow on one of the hosts. Additionally, the ‘tyranny’ of generalists is now an even bigger problem than it was with Parallel presentation because of the multiplicative advantage of the generalist over both specialists. However, if the goal is to obtain a generalist at any cost, Sequential presentation is far better than Parallel.

#### 3.2.3. Mixed Presentation

This protocol is easy to implement but offers the greatest challenge for intuition. All three phages are growing in one culture, and the success of the specialist on one host directly impacts the success of the generalist growing on that host, which in turn affects the success of the generalist on the other host, then affecting the success of the specialist on the other host. Some of this interdependence has been true with the other presentation protocols. However, with Mixed presentation, the generalist’s success can no longer be separated between host A and B: its suppression by the phage specializing on host A affects its ability to compete with the phage specializing on host B and vice versa.

There is a further complication of Mixed presentation. When an individual phage has infected one host, it cannot also be infecting another host. This means that a generalist can be at a disadvantage when infecting a poor host if good hosts are in unlimited supply [31]; however, this effect disappears as the good host declines in abundance. Thus the cost–benefit to the generalist can change over the duration of a single culture, and a quantitative understanding of the implications defies intuition.

Figure 9 shows results for Mixed presentation with high bacterial growth rate of *r* = 0.3 and fixed-count dilution of 1000. Compared to Parallel and Sequential host presentation, Mixed presentation is intermediate in favoring generalists: the generalist displaces somewhat superior specialists, but not to the degree as with Sequential presentation. The pattern for Mixed presentation with fixed-volume dilution is similar that in Figure 9 (not shown), although the generalist enjoys a somewhat greater advantage over the A-specialist.

### 3.3. Summary

To facilitate comparing the many results, Table 5 lists the key differences by protocol properties. This comparison is offered as a qualitative comparison of methods to reveal the sensitivities in outcome to seemingly subtle changes in protcol. The parameter values used (e.g., for bacterial growth rate) are not intended to represent any particular empirical system.

## 4. Discussion and Conclusions

The suite of bacterial strains and species infected by any phage tends to be narrow, at least compared to the spectrum of hosts impaired by single antibiotics. Applications of phage therapy would benefit from broad host range phages, if merely to simplify treatment and phage preparation. Fortunately, the host range of a phage is not a fixed feature, and there is a long history of ‘directing’ the evolution of phages in the laboratory to change and even expand host range. These methods for directed evolution of broad host range all expose phages to ‘new’ bacterial hosts, with the goal of evolving the phages to grow on those hosts.

Here, we considered the quantitative consequences of different protocols for the directed evolution of phage host range. We distinguished three protocols that differ in the way hosts are ‘presented’ to the evolving phage pool, all of which have been used in prior work: Parallel, Sequential, and Mixed. There were qualitative differences among these protocols in their selection of generalist versus specialist phages. Sequential presentation strongly favors generalists over specialists, Mixed presentation less so, and Parallel presentation (with an appropriate dilution protocol) provides little or no intrinsic advantage to generalists. However, the generalists evolved under Sequential presentation can be far inferior on individual hosts than the specialists they displace, so generalism can have drawbacks. Other protocol details can have modest effects. The dilution protocol should not be so extreme as to cause demographic extinction of phages.

Are there limits to the evolution of broad host ranges in phages? The works of Burrowes, Molineux and Fralick [18] and of Mapes et al. [19] suggest that host ranges can be extended across multiple strains of the same species. Mapes et al. [19] even found that expanded host-range phages could grow on strains not included in the protocol, whereas Burrowes, Molineux and Fralick [18] found the opposite. The ease of evolving expanded host range phages (and even cocktails) may well depend on the basis of bacterial resistance [2,6]. The limits to phage host ranges is a matter that looms large in the future of phage therapy.

### 4.1. Other Effects of Protocol on Evolution

Selection is not the only effect of protocol on evolution. The origin of genetic variation and stochastic loss of rare mutants are two properties to consider. These topics were not included in the numerical analyses above, but they warrant consideration in any protocol for host-range evolution.

#### 4.1.1. Recombination

Beyond the starting phage genomes, variation that is critical for evolution may be introduced during the protocol, as with mutagenesis or recombination. Studies of individual phages indicate that genetic changes are responsible for host range shifts [21,22,32,33], so methods that enhance the variation beyond the starting pool are expected to accelerate the evolution [e.g., mutagenesis, [34]].

Using three phages and a Parallel presentation protocol (i.e., the Appelmans protocol), Burrowes, Molineux and Fralick [18] evolved a broad-range *Pseudomonas* phage that was a manifold recombinant between two of the starting phages. Furthermore, they commented that minimal success at host range expansion was observed when initiating the Appelmans protocol with single phages, for which the main source of variation would have been point mutations arising at intrinsic rates. Extrapolating from these results, Burrowes, Molineux and Fralick [18] proposed that recombination greatly facilitates the evolution of broad host range in phages. An appreciation of the evolutionary benefit of recombination in phages is not without precedent: Botstein [35] proposed that evolution of phages in the wild was highly ‘modular,’ a process that necessarily relies on recombination among otherwise possibly divergent phages. Likewise, the field of directed evolution has experimentally demonstrated the value of recombination or ‘DNA shuffling’ over mutation [36].

Otherwise, recombination has rarely been argued to be important in the directed evolution of phage host range, and few protocols are designed to encourage it [although protocols to engineer host range expansion libraries rely on it, [28,29,37]]. The Appelmans protocol is an exception [18]. It has two design features that should enable high levels of natural recombination: (i) multiple starting phages, and (ii) hosts individually permissive to more than one of the initial phages. Beyond this, there are additional design properties that can influence recombination. One is the similarity among the different phages. As (homologous) recombination relies on sequence similarity, starting phages that have regions of similar sequence should be most prone to recombine, provided they can infect a common host. However, given that a recombination occurs, the phenotypic effect of that recombination will likely be greater the more that the parent sequences differ. Therefore, there may be an optimal level of sequence divergence among the phages for recombination to be useful in host range evolution. The broad host-range phage evolved by Burrowes, Molineux and Fralick [18] was a mosaic of two starting phages that were 99% similar, suggesting that even high sequence similarity does not preclude the generation of novel phenotypic variation via recombination. Yet, the evolved phage had 48 identified recombination segments (contrasted with only a single unambiguous point mutation), a number that seems extraordinarily high. It was not known how many of those exchanges were critical in the host range evolution, but the large number could reflect the intrinsically small phenotypic variation that arises via recombination when there is low genetic diversity between the parents. We are largely ignorant about how to facilitate recombination while enhancing variation for host range expansion.

For a protocol to encourage recombination, the different phages must coinfect. Co-infection requires a common host in the protocol but also requires high, approximately equal abundances of the different phages. A common host is easily employed in any of the three host-presentation protocols, but ensuring ongoing high abundances of different phages is not. Even when the protocol is deliberately initiated with equal numbers of the different phages, relative abundances will usually change rapidly because different phages will typically not have equal growth rates on common hosts. The better growing phages will thus quickly dominate the pool [38,39]. To ensure the continued opportunity for recombination, therefore, the evolving phage pool could periodically be supplemented with a stock in which all the phages are equally abundant and in which recombinants are already present. Again, this method can be employed with any of the three presentation protocols. A simple empirical method for creating a pool of recombinants is to cross streak the different phages on a plate seeded with a common host, allow phage growth, and recover the phages from the zone of overlap [40].

#### 4.1.2. Rare Mutations Are Likely to Be Lost

Virtually any protocol that amplifies phages on a permissive host will then dilute those phages before exposing them to selective conditions. This dilution has consequences for single mutants that have just arisen in the phage pool: a single mutant in the pool may not even enter the next round of cultures, much less be exposed to a host on which it can grow. For example, the Appelmans protocol generates a phage pool by combining all cultures that lyse with some cultures that do not lyse; but it then adds only a fraction of this pooled volume to the next round of cultures. In Figure 1 of Burrowes, Molineux and Fralick [18], 37 culture volumes are collected to create the phage pool, but less than 4.5 of those volumes would be used in the next round (12%). Any individual specialist that survives this bottleneck then has only a 1/8 chance of being placed in a culture with the non-permissive host on which it grows or a 1/8 chance of being placed in the culture with the permissive host (for an overall net rate of 3% that a single mutant will survive the dilution and be placed with a permissive host in the next cycle).

Other protocols may also use only an aliquot of the phage pool for the next round. In all cases, the numbers are subject at least partly to experimental control. Thus, the odds are improved under the Appelmans protocol by using fewer non-permissive hosts and by pooling fewer of the cleared wells [as per [19]]. Mixed presentations would not face the problem that a new mutant might fail to be exposed to the host on which it grows. Thus, several of these risks of mutant loss are protocol-specific. Concentrating the phage in the pool before the next round of host exposure would likely reduce loss in any protocol. Beyond these problems, there are further random processes that can lead to loss of single or even low-copy, mutant phages [41]. Of course, if host range mutations or recombinants arise at high frequency, individual losses will not matter.

### 4.2. Recommendations

We offer a few conclusions and recommendations for evolving expanded host ranges.

(1)Choose an appropriate presentation protocol. Although the protocols vary in how deferential they are to generalist phages, the utility of generalist over specialist phages will depend on the application. To take advantage of the variety of host-presentation protocols available, the protocol could be tailored to the application. For example, sequential presentation of hosts is unlikely to exist in infections (except possibly if host resistance evolves following treatment), whereas both Parallel and Mixed protocols could directly match infections. In turn, this would suggest that a Sequential protocol was not the best choice for evolving phages to treat infections. Phages evolved under Mixed presentation might be better suited to treating a mixed infection than would phages evolved under other protocols, and so on.(2)Recombination. Although much remains to understand the general importance of recombination to host range evolution, there may be little lost by encouraging recombination in a protocol. Recombination is promoted by using multiple parental phages and ensuring they are maintained at moderate frequencies. However, including different phages at the start may offer only a transient boost to recombination. The propagation of different phages on a permissive host will typically result in numerical dominance by one phage [38], so the opportunities for recombination may dissipate quickly. Periodically supplementing the phage pool with the original phage stock will maintain opportunities for recombination. Furthermore, the phage pool itself may be created with recombination so that the pool is not only supplemented with parental phages but also supplemented with recombinants among the parents. Recombination rates also depend on the sequence similarities among starting phages.(3)Early losses. If mutations are limiting, it may be beneficial to modify the protocol to reduce phage dilution and avoid phages being trapped on an inappropriate host—the latter a greater problem with Parallel presentation than, say, Mixed presentation. In the Appelmans protocol, for example, this would involve reducing the number of non-permissive hosts. Concentrating phages at the end of a cycle would reduce dilution. Alternatively, the protocol may be modified to increase the number of recombinants and mutants (as with mutagens), so that early loss is inconsequential.(4)The initial pool. A diversity in the initial pool of phages will surely increase the range of possible outcomes. As noted above, the initial pool will affect opportunities for recombination. We yet can offer little insight on whether a diverse starting pool will affect the outcome toward generalists or specialists. Starting with a single phage might seem to ensure the evolution of a generalist, but selection is agnostic to phage origins, and a single phage might well diversify into a suite of specialists.(5)Dilution. There is a risk from too high dilution in a protocol—demographic extinction of phages that would otherwise evolve. However, low dilution has its own problems. First, low dilution slows evolution merely because there is less phage growth per cycle and thus less opportunity for fitness differences to manifest. Low dilution could thus greatly increase the empirical work associated with evolving phage host range. Second, low dilution results in most of the phage growth at high multiplicity. This can facilitate the evolution of phages that are good at within-host competition at the expense of independent growth [42,43]. High dilution will usually be desirable, and any demographic effects can be offset by extending the culture duration.

## Figures and Tables

**Figure 1 antibiotics-11-01709-f001:**
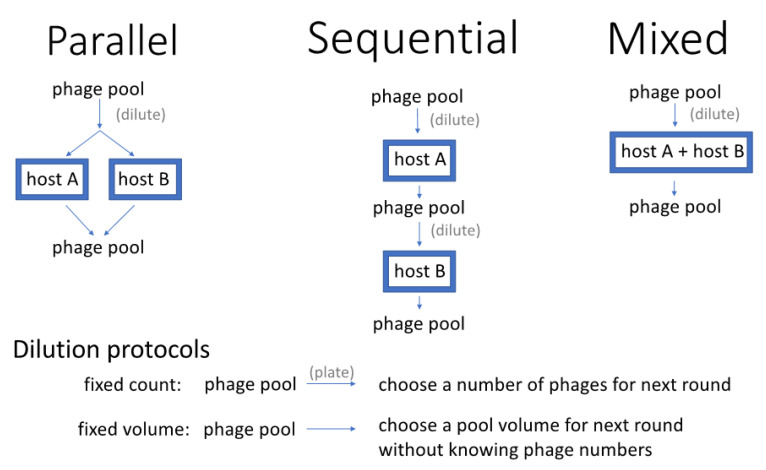
Three alternative host-presentation protocols (Parallel, Sequential, Mixed) differ in how phages are distributed to two different hosts on which growth is desired. Each box around a host type (A or B or both) represents a separate culture of hosts to which phage are added and grown. The figure illustrates a single cycle, but typically a procedure is carried out for several cycles. The figure also includes two dilution protocols that may be used when distributing phages into the next cycle (fixed count, fixed volume).

**Figure 2 antibiotics-11-01709-f002:**
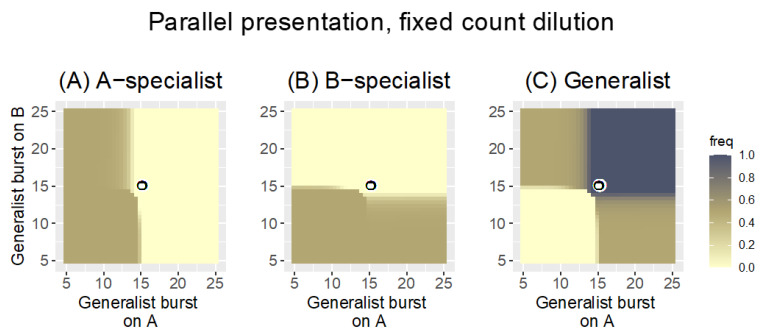
Heat maps of Parallel presentation with equal burst sizes of the two specialists (*b_A|A_* = 15, *b_B|B_* = 15, as indicated by the coordinates of the black-on-white ring). The three panels show the frequency of the A-specialist (**A**), the B-specialist (**B**), and the generalist (**C**), respectively, across a range of generalist burst sizes. As given in the key, phage frequency at 1000 time steps is indicated by color, with darker colors representing higher frequencies. The generalist is lost if both its burst sizes are less than those of the specialists. Each specialist is lost when its burst size falls below that of the generalist on that host. There are substantial zones in which two phages are maintained (either the two specialists are maintained or a specialist and the generalist are maintained), although only the generalist is maintained in the upper right quadrant. Because phages with the highest burst sizes on each host are maintained, Parallel presentation is a ‘fair’ arbiter of phage growth. There is a slight advantage to being a generalist in that the generalist displaces the specialist even when the generalist bursts are both slightly less than that of the respective specialist. Bacterial growth rate is *r* = 0.3 and the fixed-count dilution reduced phage density to 1000 every 20 time steps. Phage frequencies were calculated as a phage’s density relative to the combined density of all phages.

**Figure 3 antibiotics-11-01709-f003:**
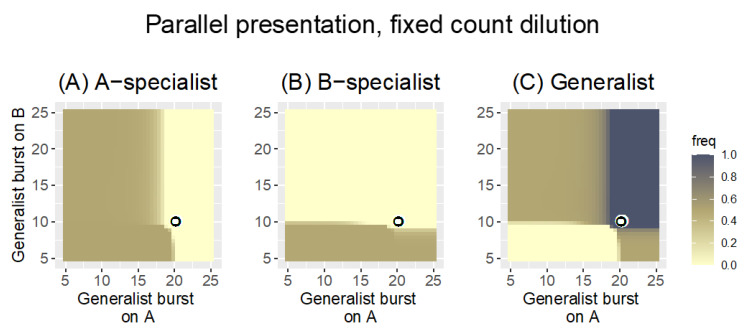
Heat maps of Parallel presentation with unequal burst sizes of the two specialists (*b_A|A_* = 20, *b_B|B_* = 10, as indicated by the coordinates of the black-on-white ring). The three panels show the frequency of the A-specialist (**A**), the B-specialist (**B**), and the generalist (**C**). As in Figure 2, the generalist is maintained on either or both hosts according to whether its burst size exceeds that of the specialist on that host, and there are large zones in which two phages are maintained. Each panel shows, for the phage indicated in the title, its frequencies at time 1000 based on generalist and specialist burst sizes. Each specialist is lost when the generalist burst size exceeds that of the specialist on that host. The generalist is lost only when its burst sizes are less than those of the respective specialists on both hosts. Again, Parallel presentation is a ‘fair’ arbiter of phage retention, and again, there is a slight advantage to being a generalist in that the generalist displaces the specialist even when the generalist bursts are each slightly less than that of the respective specialist. There is no obvious ‘tyranny’ of generalists on the good host here. However, a tyranny is not necessarily expected because the bacterial growth rate is so high (0.3) that the poor host (B) reaches a high density before the poor phage overwhelms it—resulting in a large *N_B|B_* despite its small burst. Bacterial growth rate is 0.3 and dilution reduces phage density to 1000 every 20 time steps. The key is the same as in Figure 2.

**Figure 4 antibiotics-11-01709-f004:**
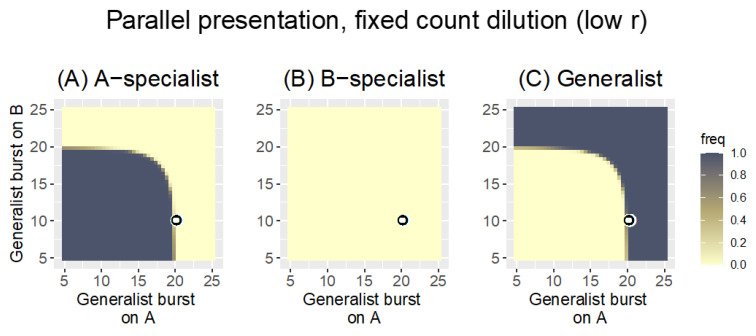
Heat maps of Parallel presentation with unequal burst sizes of the two specialists (*b_A|A_* = 20, *b_B|B_* = 10, as indicated by the black-on-white ring). The three panels show the frequency of the A-specialist (**A**), the B-specialist (**B**), and the generalist (**C**). The trial conditions are the same as in Figure 3 except that the bacterial growth rate has been lowered to *r* = 0.1. With this change, the pattern is radically different from before (Figure 3). Now, only one phage is ever maintained, and the B-specialist is lost everywhere. Although the specialist burst sizes are unequal, the pattern of phage maintenance is symmetric, the A-specialist being the only phage maintained up to the point that the generalist burst on either host exceeds 20 (20 is the burst size of the A-specialist). As in Figure 3, dilution reduces phage density to 1000 every 20 time steps. The key is the same as in Figure 2 and Figure 3.

**Figure 5 antibiotics-11-01709-f005:**
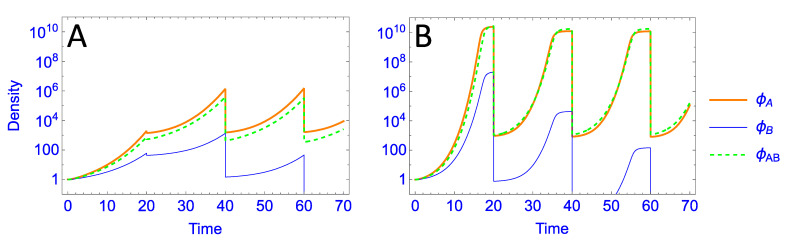
Temporal, short-term dynamics of phage in Parallel-presentation trials of Figure 4 (panel **A** above) and Figure 3 (panel **B** above); the generalist has burst sizes of 17 on both hosts, representing a point in the heat map of Figure 4 at which only the A-specialist is maintained but at which the A-specialist and generalist are maintained in Figure 3. Comparison of these dynamics reveals why the two growth conditions yield such differences in patterns of phage maintenance between the different conditions of Figure 3 and Figure 4. With a low bacterial growth rate (*r* = 0.1, **A**) the phages never exhaust the bacteria before dilution; in essence, the phages are maintained in a constant state of exponential growth, and the fastest grower sets the threshold for dilution such that other phages are progressively diluted to extinction. Increasing the bacterial growth rate to *r* = 0.3 (**B**) has the unintuitive effect of allowing accelerated phage growth toward the end of a cycle; as cell densities reach high levels, phage growth accelerates, exhausting the cells. Even slowly growing phages can recover from low transfer densities and attain high densities by cycle’s end. This exhaustion of hosts allows slowly growing phages on one host to be maintained along with fast-growing phages on the other host. Note that there are no units of density or time because they are relative to each other, to the other variables, and to the parameters. See Methods for an explanation.

**Figure 6 antibiotics-11-01709-f006:**
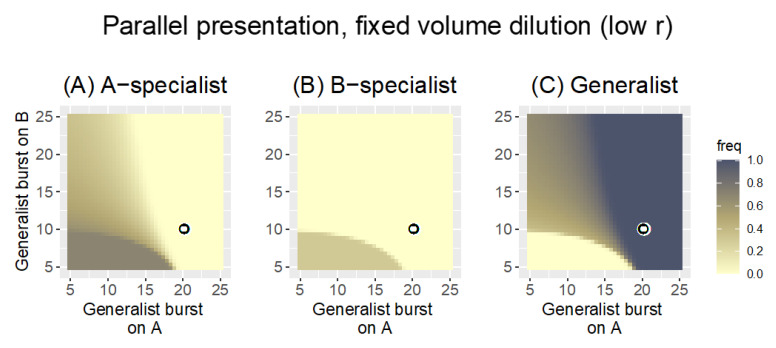
Parallel presentation with fixed volume dilution and low bacterial growth rate (*r* = 0.1). The three panels show the frequency of the A-specialist (**A**), the B-specialist (**B**), and the generalist (**C**). Trials are the same as detailed in Figure 4 except that more phage are transferred every cycle: here, 5% of the phage pool at the end of a cycle is added to each new culture instead of 1000 phage. Figure 5 showed that total phage density reached just over 10^6^ per cycle with fixed-count dilution; with a final density of 10^6,^ the fixed volume dilution here would transfer 50,000 phage instead of 1000. The final density here at time 1000 ultimately exceeds 10^6^ by orders of magnitude, however. The pattern is now much closer to that of Figure 3 (*r* = 0.3, fixed-count dilution) than that of Figure 4: two phages are maintained over much of the space. Yet, there are also differences from the patterns in these other two cases—the generalist is maintained over a wider set of burst size values than in Figure 3. Burst sizes of the two specialists are again given by the coordinates of the black-on-white ring. The key is the same as in Figure 2 and Figure 3.

**Figure 7 antibiotics-11-01709-f007:**
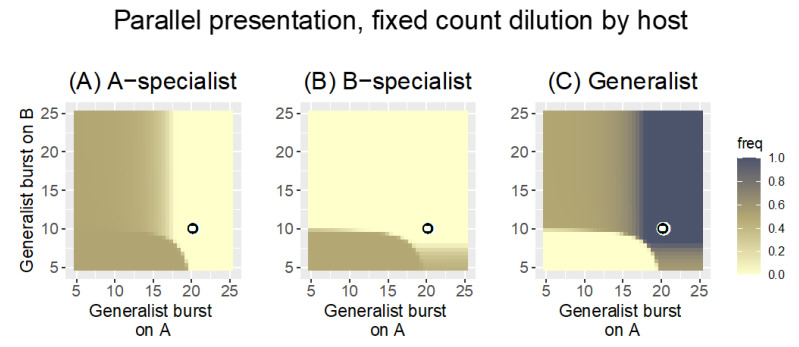
Trials of Parallel presentation under the same conditions as in Figure 4 (bacterial growth rate *r* = 0.1, fixed-count dilution), except that the phage pool from each host is diluted to density 1000 in each culture separately before being combined into a common pool. The three panels show the frequency of the of the A-specialist (**A**), the B-specialist (**B**), and the generalist (**C**) after 1000 time steps. In contrast to the pattern with fixed-count dilution and low bacterial growth in Figure 4, the pattern with separate dilutions is now similar to that with high bacterial growth rate (*r* = 0.3, Figure 3), although the generalist enjoys somewhat more of an advantage here than in Figure 3. Burst sizes of the two specialists are again given by the coordinates of the black-on-white ring. The key is the same as in Figure 2 and Figure 3.

**Figure 8 antibiotics-11-01709-f008:**
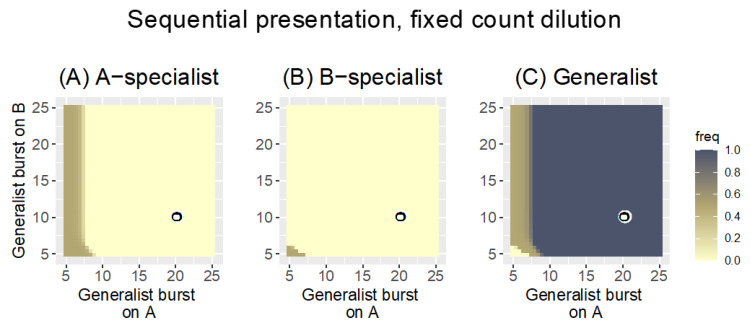
Sequential host presentation provides a huge advantage for the generalist. The three panels show the frequency of the of the A-specialist (**A**), the B-specialist (**B**), and the generalist (**C**) after 1000 time steps. Trials were run under the same conditions as in Figure 3 (*r* = 0.3, fixed-count dilution), but hosts were presented sequentially (alternated). The generalist is lost in only a tiny corner of the space, and both specialists are lost in most of the space. Furthermore, the generalist displaces both specialists throughout much of the space in which one or both specialists are far superior to the generalist on their respective hosts. Burst sizes of the two specialists are again given by the coordinates of the black-on-white ring. The key is the same as in Figure 2 and Figure 3.

**Figure 9 antibiotics-11-01709-f009:**
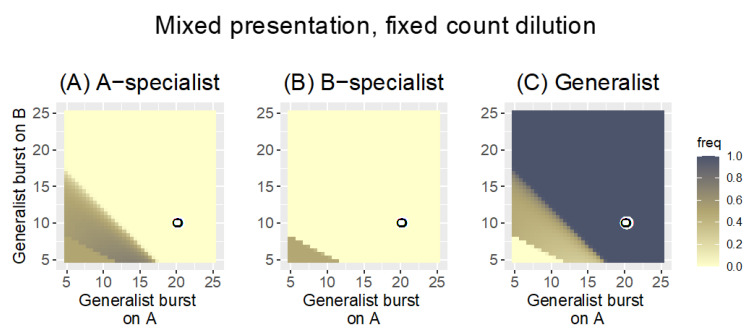
Mixed presentation with fixed-count dilution (to 1000 phage) and high bacterial growth (*r* = 0.3). The three panels show the frequency of the of the A-specialist (**A**), the B-specialist (**B**), and the generalist (**C**). The outcome is intermediate between that of Figure 3 (Parallel) and Figure 8 (Sequential) in that the generalist displaces both specialists in much of the space where one or both specialists have higher burst sizes. Burst sizes of the two specialists are again given by the coordinates of the black-on-white ring. The key is the same as in Figure 2 and Figure 3.

**Table 1 antibiotics-11-01709-t001:** Notation used in equations.

Notation	Meaning	Values Used
$P_{i|j}$	density of phage *i* growing on host *j* (function of time)	> 0
$C_{j}$	density of host *j* (function of time)	> 0
$I_{ij}$	density of host j infected with phage *i* (function of time)	> 0
$\lambda_{ij}$	lysis rate of $I_{ij}$	1.0
$b_{i|j}$	number of phage progeny released at lysis from an $I_{ij}$ infection	5–25
$k_{ij}$	adsorption rate constant of phage *i* onto cells of strain *j*	10^−9^
*r* _ *j* _	Growth rate of bacterial strain *j*	0.1, 0.3

**Table 2 antibiotics-11-01709-t002:** Host-specific growth of different phages per culture.

		Host A	Host B
Phage	Φ_A_	*N_A|A_*	1
	Φ_B_	1	*N_B|B_*
	Φ_AB_	*N_AB|A_*	*N_AB|B_*

**Table 3 antibiotics-11-01709-t003:** Combined growth after pooling with Parallel presentation.

Phage	Descendants per Cycle
Φ_A_	*N_A|A_* +1
Φ_B_	1 + *N_B|B_*
Φ_AB_	*N_AB|A_* + *N_AB|B_*

**Table 4 antibiotics-11-01709-t004:** Phage outputs during a cycle of Sequential presentation.

Phage	Descendants per Cycle
Φ_A_	*N_A|A_* · 1
Φ_B_	1 · *N_B|B_*
Φ_AB_	*N_AB|A_* · *N_AB|B_*

**Table 5 antibiotics-11-01709-t005:** Summary of Results.

Protocol	Dilution	Bacterial *r*	Outcome
Parallel	fixed count	0.3	Phages with the best growth on a host are retained; by and large, generalists have no specific advantage over specialists.
“	fixed count	0.1	With the low bacterial growth rate, the fixed count dilution allows all phages growing on the poorer host to be lost; only one phage is maintained in the pool.
“	fixed volume	0.1	Despite the low bacterial growth rate, the change in dilution protocol now retains phages for both hosts; the generalist has a modest advantage over specialists
“	fixed count by host	0.1	The change in dilution protocol ensures that phages are retained for both hosts; the generalist has a slight advantage over specialists, but the method broadly rewards phage growth on a per-host basis
Sequential	fixed count	0.3	Generalists displace specialists except when the specialist is extremely superior
Mixed	fixed count	0.3	Generalists are strongly favored over specialists; the outcome is intermediate between Parallel and Sequential presentation

## Data Availability

C code files, R code files, and Mathematica files are available at https://github.com/bull71/host_range (accessed on 24 November 2022).

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
