# Peer review of "Modeling the Directed Evolution of Broad Host Range Phages"

_antibiotics, 2022, doi:10.3390/antibiotics11121709_

Round 1
Reviewer 1 Report
Bull et al have worked on employing varied analytical tools to compare different methods of selecting and screening phages with different bacterial hosts. This comparative analysis seems to be the first of its kind that looks at the different methods of directed evolution of phages in a standardized fashion with great attention to detail. The accuracy of the theoretical analysis was remarkable in corelating the selection protocol with the preference towards generating specialist or generalist phage. The investigation of the effect of dilution protocols on the kind of phage generated was an additional nice aspect.
Overall it is well written manuscript going on every data in much details.
Page 16 Line 599 – It would be great if the authors could rephrase this sentence to improve the clarity of the thought.
Author Response
R1: Bull et al have worked on employing varied analytical tools to compare different methods of selecting and screening phages with different bacterial hosts. This comparative analysis seems to be the first of its kind that looks at the different methods of directed evolution of phages in a standardized fashion with great attention to detail. The accuracy of the theoretical analysis was remarkable in corelating the selection protocol with the preference towards generating specialist or generalist phage. The investigation of the effect of dilution protocols on the kind of phage generated was an additional nice aspect.
Overall it is well written manuscript going on every data in much details.
Page 16 Line 599 – It would be great if the authors could rephrase this sentence to improve the clarity of the thought
Rewritten to clarify
Reviewer 2 Report
The manuscript with tittle ”Modelling the directed evolution of broad host range phages” written by James J. Bull et al. Compares different methodologies to obtain generalist phages from two different specialist phages by mathematic model approaches based on differential equations and both a more simple heuristic model and a numerical analysis which provides more information about how the different phage presentations increase phage host range. The authors compared three different methodologies, parallel presentation, mixed presentation and sequential presentation. Within the first type of presentation several variations were performed in order to optimize the model. Authors provide data for phage host range increase based on two initial phages and two different bacterial strains, however they claim that this model could be applicable to further phage and bacterial strain numbers The authors determined that models had different outcomes and that the use of one or other methodology may depend on the final goal of phages breeders.
As the authors described in the introduction new alternatives to treat diseases are needed other than antibiotics due to the alarming increases of antibiotic resistant bacteria. Further research is needed in bacteriophage treatments, interaction and breeding and this manuscript provides valuable information for both researchers and medical scientists to find phages having wider host range that can be used regularly for treating diseases or injuries.
This paper well describes the problematic in the introduction justifying the need of the research performed and properly describing the different methodologies or presentations. I believe that a figure showing all three presentation types will help the reader to follow the paper, in this figure also the different assumptions and conditions described in the manuscript can be summarized.
The methodology describes well the design of the experiments as well as all the factors that were considered for the mathematical model. However, they should be clearer explaining the different dilution protocols and bacterial concentration used. Although having explanations and justification of the decision taken in the protocols the core and important points from the experiment get lost in the reading and is difficult to follow where is the key information. Same occurs in the equation parameters descriptions; a table should help to follow the equations description.
The results are very difficult to follow and again the reader get easily lost in all the information provided, this part is very dense with lot of variation and suggestions and really need to be well explained and summarized and maybe accompanied with figures that facilitate the reading. For example the information obtained from the heuristic method is not further discussed in the paper and might be moved to supplemental results.
Since figure comparisons are made it would be easy to the reader to have them in the same page and figure, a good description of what the graphs represent is needed because it might lead to misunderstanding of the results.
For parallel presentation and all the analysis performed at least on this model a table summing up characteristics and results should be provided. This section might need to be better organized and very well explained, maybe reducing the text and leading the reader to the key facts of each presentation.
For the point 3.3, since no results are provided coming from the model, only assumptions can be made, this need to be included into the discussion.
Discussion needs to be rewritten or more results need to be provided for recombination and lost mutations and phages, a conclusion cannot be made without data.
It would be nice to compare the in silico results you obtained with the biological system, will you expect the same results? As mentioned in the manuscript all the assumptions taken in the model will lead to variable results in the laboratory.
Why not show in the paper the model with high number of bacterial strains that will provide a bigger host range, if the model is applicable to this system the results should be shown. The final goal is to generate a wide host range phage able to infect different types of bacteria.
Some minor comments:
Line 35-36: this is not easy to understand
Lines 42-45: this part needs to be better explained since is a key part of the introduction, that will justify the paper.
Line 50: even instead of ever
Line 49-52 is there any publication showing that bacteria evolve resistant to phages? In that case please cite the study.
Line 55:” -the use a laboratory…”, a preposition is missing there
Line 63: …”and the user may prefer generalist phages to specialist” is already said, this sentence can be omitted, or at least rewritten to give more importance to the paper.
Line 69: Remove the “the” and make hosts singular
As mentioned above, results and M&M sections need to be better and more clearly described.
Author Response
R2: The manuscript with tittle ”Modelling the directed evolution of broad host range phages” written by James J. Bull et al. Compares different methodologies to obtain generalist phages from two different specialist phages by mathematic model approaches based on differential equations and both a more simple heuristic model and a numerical analysis which provides more information about how the different phage presentations increase phage host range. The authors compared three different methodologies, parallel presentation, mixed presentation and sequential presentation. Within the first type of presentation several variations were performed in order to optimize the model. Authors provide data for phage host range increase based on two initial phages and two different bacterial strains, however they claim that this model could be applicable to further phage and bacterial strain numbers The authors determined that models had different outcomes and that the use of one or other methodology may depend on the final goal of phages breeders.
As the authors described in the introduction new alternatives to treat diseases are needed other than antibiotics due to the alarming increases of antibiotic resistant bacteria. Further research is needed in bacteriophage treatments, interaction and breeding and this manuscript provides valuable information for both researchers and medical scientists to find phages having wider host range that can be used regularly for treating diseases or injuries.
This paper well describes the problematic in the introduction justifying the need of the research performed and properly describing the different methodologies or presentations. I believe that a figure showing all three presentation types will help the reader to follow the paper, in this figure also the different assumptions and conditions described in the manuscript can be summarized.
Seems like a good idea, and Rev 3 thought so also. We added such a figure (now Fig. 1)
The methodology describes well the design of the experiments as well as all the factors that were considered for the mathematical model. However, they should be clearer explaining the different dilution protocols and bacterial concentration used. Although having explanations and justification of the decision taken in the protocols the core and important points from the experiment get lost in the reading and is difficult to follow where is the key information. Same occurs in the equation parameters descriptions; a table should help to follow the equations description.
We usually do include such tables. Not sure why we did not, but we’ve now added one. Good call.
The results are very difficult to follow and again the reader get easily lost in all the information provided, this part is very dense with lot of variation and suggestions and really need to be well explained and summarized and maybe accompanied with figures that facilitate the reading. For example the information obtained from the heuristic method is not further discussed in the paper and might be moved to supplemental results.
For sure, there is a lot contained in this study. The Methods are about as succinct as we can make them, and it is unlikely that anyone not already familiar with the type of models we use will want the extra text; Methods are self-contained and complete. For Results, have done our best to focus on the main points. But the problem studied here is intrinsically difficult because the protocol affects evolution AND demography, and protocol details (such as dilution) affect both. We could merely skip parameter zones in which demography has an effect separate of evolution, but part of our goal is to have the reader realize that the protocol does indeed have these disparate effects. Nonetheless, we did make several edits in Results to improve clarity.
When presenting numerical results, we did indeed make references to the heuristics, but they were perhaps too cryptic. We have made them more overt. We feel strongly about retaining them because they greatly shaped OUR intuition. Indeed, to us, the numerical results seem almost secondary to the heuristics (except, of course, for Mixed infections because we have no heuristic for Mixed). But our esteem for the heuristic results may not have been obvious the way the paper was written, so we tried to make that clearer. We are glad to be notified that this wasn’t clear, because the heuristic results are so important to us.
Since figure comparisons are made it would be easy to the reader to have them in the same page and figure, a good description of what the graphs represent is needed because it might lead to misunderstanding of the results.
Upon getting this review, we considered this suggestion at length. Yes, the different analyses are difficult to assimilate, and something needed to be done. Our response has been to add a table (5) at end of Results that summarizes the different results. A figure combining all current figures would face several problems: (i) the legend would span 2 pages or more (even with the elimination of redundant text), and such a legend would be incomprehensible because of the many different components; (ii) the figure would be confusing when presenting the first few sets of results before the others because the reader would not know to ignore most of the panels; it’s only after all the results have been presented that a combined figure would help; (iii) to be understandable, such a figure would need separate headings for each panel, in essence just packaging into one space what we already have in our separate figures. We thus opted for a table as a summary. The table is far more practical.
For parallel presentation and all the analysis performed at least on this model a table summing up characteristics and results should be provided. This section might need to be better organized and very well explained, maybe reducing the text and leading the reader to the key facts of each presentation.
We have added such a table (5) albeit that the table also includes Sequential and Mixed. While we agree that the flow of outcomes under Parallel is complicated, we think this difficulty is largely unavoidable. The complexities stem from an interaction between bacterial growth and the dilution protocol, and the effect comes from demography rather than evolution. There is just a lot going on.
For the point 3.3, since no results are provided coming from the model, only assumptions can be made, this need to be included into the discussion.
Even before submission, we were torn about where to put this section. We fully agree with the reviewer that that the section really does not constitute Results. We have moved it to Discussion.
Discussion needs to be rewritten or more results need to be provided for recombination and lost mutations and phages, a conclusion cannot be made without data.
This comment overlaps with the previous one. Now that we have moved the section into Discussion, the lack of original analyses should no longer be an issue – we are merely discussing prior work (with a new conclusion of our own).
It would be nice to compare the in silico results you obtained with the biological system, will you expect the same results? As mentioned in the manuscript all the assumptions taken in the model will lead to variable results in the laboratory.
Testing these models is of course the next step. This is a models paper, the purpose of which is to lay a foundation that may or may not hold up empirically. Our work is intended to generate hypotheses. To make this clearer, our revision has added a bit of language to that effect. Down the road, we do expect to conduct some of these tests, but the paper is about models and is already longer than most papers.
Why not show in the paper the model with high number of bacterial strains that will provide a bigger host range, if the model is applicable to this system the results should be shown. The final goal is to generate a wide host range phage able to infect different types of bacteria.
Yes, we pondered this expansion before submission. It is a worthy question, but the analysis is far from trivial. With just 3 hosts instead of 2, there are 7 possible phage types: three specialists, one 3-host generalist, and three different 2-host generalists. That translates into 12 burst size parameters. For comparison, each of our data figures depicts 3 phages and 4 burst size parameters. The analysis of 3 hosts is so challenging that it would be sufficient for its own paper.
Some minor comments:
Line 35-36: this is not easy to understand Modified
Lines 42-45: this part needs to be better explained since is a key part of the introduction, that will justify the paper.
We have rewritten
Line 50: even instead of ever Actually, we did indeed intend ‘ever’ but maybe it needs a hyphen? (ever-larger??). Maybe this is language from the past.
Line 49-52 is there any publication showing that bacteria evolve resistant to phages? In that case please cite the study.
A few phage therapy citations added. It is very well known that bacteria evolve resistance to phages in vitro.
Line 55:” -the use a laboratory…”, a preposition is missing there Good catch.
Line 63: …”and the user may prefer generalist phages to specialist” is already said, this sentence can be omitted, or at least rewritten to give more importance to the paper.
Phrase chopped.
Line 69: Remove the “the” and make hosts singular Changed.
As mentioned above, results and M&M sections need to be better and more clearly described.
We are hoping that Table 5 fixes any problem with Results. Methods are complete and explanatory for modelers.
Reviewer 3 Report
The authors have used a combination of simple analytical methods and numerical analyses to study the effect of different protocols on the selection of generalist versus specialist phages. They observed that three presentation protocols have profoundly different consequences for the evolution of generalist versus specialist phages. Sequential presentation favors generalist almost to the exclusion of specialists, whereas Parallel presentation does the least so. However, other protocol attributes also have effects on selection and phage maintenance. It is also noted that protocols can be designed to enhance recombination to augment evolution and reduce stochastic loss of newly-arisen mutants. This study can be very useful as antimicrobial resistance is on the rise. The study is thorough and well-designed. However, there are a few points that should be addressed by the authors.
- Authors should cite the proportion data in the introduction on the importance of phage treatment vs. antibiotics.
- It would be ideal if the authors could design a figure in the introduction to show the difference between parallel, sequential, and mixed presentations.
- Authors should try to improve the quality of the figures.
- The authors discussed the earlier studies in the last paragraph of the discussions (lines 721-728). Authors should their conclusions win details.
- Lines, 485-489. Authors should simplify the language.
- Figure 4, the unit of time and density should be there in the graph.
- There are multiple grammatical errors in the manuscript. Authors should go through it thoroughly and try to improve.
Author Response
R3: The authors have used a combination of simple analytical methods and numerical analyses to study the effect of different protocols on the selection of generalist versus specialist phages. They observed that three presentation protocols have profoundly different consequences for the evolution of generalist versus specialist phages. Sequential presentation favors generalist almost to the exclusion of specialists, whereas Parallel presentation does the least so. However, other protocol attributes also have effects on selection and phage maintenance. It is also noted that protocols can be designed to enhance recombination to augment evolution and reduce stochastic loss of newly-arisen mutants. This study can be very useful as antimicrobial resistance is on the rise. The study is thorough and well-designed. However, there are a few points that should be addressed by the authors.
- Authors should cite the proportion data in the introduction on the importance of phage treatment vs. antibiotics.
The submitted draft did indeed omit references explaining the social relevance of phage therapy but now include some.
2. It would be ideal if the authors could design a figure in the introduction to show the difference between parallel, sequential, and mixed presentations.
So done. We should have thought of this ourselves.
3. Authors should try to improve the quality of the figures.
The figures may give the appearance of low resolution because these are tiled graphs, and the tiles are discrete. The figure resolution is 300 dpi, but each tile is much larger than a ‘dot.’ There are ~1600 separate tiles per panel, and the colors of each tile are assigned according to the numerical trial at that coordinate – no blending. To us, this is superior to a graphics method that artificially creates a color transition. We could somewhat reduce the tile size by iterating smaller increments in burst sizes (and thus longer computational runs), but for the focus of this study, there is little gained. We are only interested in the broadest of patterns, and the figures capture those patterns.
4. The authors discussed the earlier studies in the last paragraph of the discussions (lines 721-728). Authors should their conclusions win details.
We are not entirely sure what the objection is to this paragraph (which is no longer the final paragraph of the paper). It’s simply a comment on the limits of evolution and that the mechanisms by which bacteria block phages could matter to the outcome.
5. Lines, 485-489. Authors should simplify the language.
Rewritten
6. Figure 4, the unit of time and density should be there in the graph.
As explained in Methods, the units are relative. All that matters in Fig. 4 is that the same units would apply to the right and left panels (which they do). However, one line has been added to the caption.
7. There are multiple grammatical errors in the manuscript. Authors should go through it thoroughly and try to improve.
We have attempted to do so.
Round 2
Reviewer 2 Report
I would like to thank you the authors for all paper modifications and responses to my comments, especially in such a short time!
I think these modification makes the manuscript easy to follow and to understand.
Minor comments:
Line 97 check spelling desirved
Line 107-108 rewrite The host presentation protocols instead propagate phages agnostically, before it is known whether any phages exist that grow on the non-permissive hosts
Line 434, add a comma after now: (Fig. 3). Now, only one p…
Figure 5; A and B panels are missing
Author Response
All requested changes have been made. The level of competence in this review is impressive.